# Dietary Postbiotics Reduce Cytotoxicity and Inflammation Induced by Crystalline Silica in an In Vitro RAW 264.7 Macrophage Model

**DOI:** 10.3390/foods11060877

**Published:** 2022-03-19

**Authors:** Xue Du, Jessica Rodriguez, Josephine Wee

**Affiliations:** Department of Food Science, The Pennsylvania State University, State College, PA 16802, USA; xue.du.snow@gmail.com (X.D.); jr1617@rutgers.edu (J.R.)

**Keywords:** postbiotics, silica, macrophage, NLRP3 inflammasome, cytotoxicity, IL-1β, IL-1α

## Abstract

Crystalline silica (cSiO_2_) particles are naturally existing environmental toxicants. Exposure to cSiO_2_ could cause local or systemic inflammation and aggregate inflammation-associated diseases. Dietary postbiotics are reported to possess anti-inflammatory activities; however, their effects on cSiO_2_-triggered inflammation are unknown. Here, we investigate the impact of postbiotics from *Lacticaseibacillus rhamnosus* (LGG), *Limosilactobacillus reuteri* (L.reu), and *Bifidobacterium animalis* subsp. *lactis* Bb12 (BB12) on cSiO_2_-induced cytotoxicity and IL-1 cytokines in vitro using macrophages. The postbiotics used in this study were cell-free fractions of a probiotic growth medium collected at different time points. The in vitro model used was the wild-type murine macrophage RAW 264.7 cell line stably transfected with the inflammasome adapter protein, ASC. Our results indicate that all the postbiotics could reduce cSiO_2_-induced cytotoxicity in the wild-type and ASC macrophages and the effects were OD-dependent. Following priming with a lipopolysaccharide, cSiO_2_ treatment resulted in robust inflammasome activation in ASC, as reflected by the IL-1β release. These responses were minimal or absent in the wild-type RAW cells. All the postbiotics decreased the release of IL-1β from ASC; however, only LGG and BB12 reduced the IL-1β secretion from wild-type cells. Only the L.reu postbiotics reduced the IL-1α release from ASC. We conclude that the postbiotics from LGG, BB12, and L.reu can protect macrophages against cSiO_2_-induced cytotoxicity and suppress IL-1β activation.

## 1. Introduction

The term “postbiotics” emerged in recent years and was newly defined by the International Science Association of Probiotics and Prebiotics (ISAPP) in September 2021 as “a preparation of inanimate microorganisms and/or their components that confers a health benefit on the host” [1]. Postbiotics were previously defined as “any factor resulting from the metabolic activity of a probiotic or any released molecule capable of conferring beneficial effects to the host in a direct or indirect way” [2]. Though the new definition by the ISAPP categorizes inactivated microorganisms as postbiotics, most studies on postbiotics focus on the metabolites from probiotic microorganisms, not the inactivated microorganisms [3]. Postbiotics can be categorized by their elemental composition, including lipids (such as short-chain fatty acids), proteins (including bioactive peptides and enzymes), carbohydrates (such as exopolysaccharides), vitamins, organic acids, and complex molecules (such as lipoteichoic acids) [4]. In most cases, postbiotics are considered as a set of microbial molecules rather than single metabolites; however, published studies lack a functional characterization of these metabolites [5]. Similar to probiotics, the consumption of adequate amounts of postbiotics could ameliorate chronic inflammation and, consequently, exhibits beneficial effects toward metabolic- and inflammation-related diseases, such as obesity, diabetes, Alzheimer’s disease, and autoimmune diseases (such as systemic lupus erythematosus) [6,7,8,9]. For example, postbiotics derived from *Faecalibacterium prausnitzii* decreased the synthesis of the proinflammatory tumor necrosis factor-α (TNF-α) and interleukin-12 (IL-12) cytokines, and increased the secretion of the anti-inflammatory interleukin-10 (IL-10) cytokine in trinitrobenzenesulfonic acid-induced colitis mice in a Crohn’s disease model [10]. Compared to probiotics, postbiotics are considered to be a safer choice for premature infants, immunocompromised patients, and the elderly since the direct use of live microorganisms may induce bacteria translocation and infection, and prevent the normal colonization of other microbiota [11,12,13]. In addition, supplementation with probiotics may aggravate pre-existing severe intestinal inflammation in patients with inflammatory bowel disease [2,11]. Postbiotics do not contain viable microorganisms and, thus, eliminate such risks [14]. In addition, postbiotics are better absorbed, metabolized, distributed, and excreted, and may be more bioactive compared to probiotics [15]. Thus, postbiotics, instead of probiotics, are the main interest in this study.

Crystalline silica (cSiO_2_) are naturally existing particles that are environmental toxicants to humans and animals [16]. Almost every mineral deposit contains cSiO_2_; thus, workers in heavy construction industries are exposed to high amounts of cSiO_2_, for example, individuals working in masonry and plastering, heavy construction, painting and paper hanging, iron and steel foundries, and metal services [17]. Occupational exposure to cSiO_2_ is linked to chronic inflammation and pathogenesis of autoimmune diseases, such as systemic lupus erythematosus, rheumatoid arthritis, and scleroderma [18,19,20,21]. Although the anti-inflammatory effects of postbiotics have been demonstrated against pathogen-associated triggering in in vitro models [22,23], the impact of postbiotics on environmental toxicant-induced inflammation is not known. We previously established an in vitro model using RAW 264.7 macrophages to study the effect of the omega-3 polyunsaturated fatty acid, docosahexaenoic acid (DHA), on cSiO_2_-induced inflammation and demonstrated that DHA can reduce proinflammatory interleukin-1 (IL-1) cytokine secretion [24]. Subsequently, we concluded that the release of pro-inflammatory interleukin-1β (IL-1β) cytokines in macrophages upon cSiO_2_ introduction requires two steps: (1) a priming step where the nuclear factor kappa B (NF-κB) is activated by priming agents, such as lipopolysaccharides (LPS), and produces pro-IL-1β; and (2) a second step where the NACHT, LRR, and PYD domains containing protein 3 (NLRP3) inflammasome are assembled upon cSiO_2_ stimulation and caspase-1 is activated to cleave pro-IL-1β to mature IL-1β.

Here, we studied the effects of the postbiotics produced by *Lacticasebacillus rhamnosus* (LGG), *Limosilactobacillus. reuteri* (L.reu) and *Bifidobacterium animalis* subsp. *lactis* Bb12 (BB12) on cSiO_2_-induced cytotoxicity and the IL-1 release on the wild-type RAW 264.7 macrophages (WT). We further compared cSiO_2_-induced cytotoxicity and the pro-inflammatory cytokine release of WT with RAW-ASC macrophages (ASC), which are WT macrophages stably transfected with the apoptosis-associated speck-like protein containing a CARD domain. This protein is critical for NLRP3 inflammasome assembly and activation and is necessary for cSiO_2_-induced inflammation [25]. We hypothesize that the postbiotics from LGG, L.reu, and BB12 could decrease cSiO_2_-induced cytotoxicity and the pro-inflammatory IL-1 cytokine release in the WT and ASC macrophages. To our knowledge, this is the first study that utilizes the RAW-ASC model that investigates the effect of dietary postbiotics on cSiO_2_-induced toxicity and inflammation in vitro.

## 2. Materials and Methods

### 2.1. Experimental Design

A summary of the experimental timeline and conditions is shown in Figure 1. The goal of this work is to investigate the impact of postbiotics obtained from *Lacticasebacillus rhamnosus* ATCC 53102 (LGG), *Limosilactobacillus reuteri* ATCC 23272 (L.reu), and *Bifidobacterium animalis* subsp. *lactis* Bb12 (BB12) on cSiO_2_-stimulated macrophages. To achieve this, the cytotoxicity of the postbiotics collected at different ODs was evaluated, firstly to select postbiotics that would not affect the viability of macrophages and, secondly, to rule out the changes of cell numbers caused by postbiotics before cSiO_2_ stimulation. For this purpose, wild-type RAW 264.7 (WT) and RAW-ASC (ASC), which are WT-transfected with the fusion CFP-ASC protein [24], were seeded at 1 × 10^5^ cells/well in 24-well plates in 1 mL of RPMI. The cells were cultured in a complete medium (a phenol red-free RPMI medium with 10% FBS and 1% penicillin–streptomycin) for 24 h, washed three times with DPBS, and the medium was replaced with 1 mL of a serum-deprived medium (a phenol red-free RPMI medium with 0.25% FBS and 1% penicillin–streptomycin) [24] to avoid potential interferences of the FBS with endpoint cytokine measurements [26]. The cells were treated with either 20 µL of postbiotics collected from OD 1 to 5 or MRS (as control) for another 24 h, and supernatants were collected for a lactate dehydrogenase (LDH) assay. Prior to conducting the experiment, we measured the pH of mixtures containing 40 µL of each postbiotic and 2 mL of a serum-deprived medium (same ratio of postbiotics to medium as used in the experiment) to rule out the potential impacts of pH change on the cell viability. The pH of the individual mixture was 7.3 (±0.1), which was the same as the serum-deprived medium alone.

To evaluate the effect of postbiotics on cSiO_2_-stimulated macrophages, the cells were seeded at 1 × 10^5^ cells/well in 24-well plates and incubated in a complete medium for 24 h. The cells were washed with DPBS and the medium was switched to a serum-deprived medium (1 mL/well) before the addition of the MRS (control) or postbiotics (20 µL/well). After incubation for another 24 h, the cells were washed with DPBS and primed with LPS (Sigma Aldrich, St, Louis, MO, USA) by replacing the media with 1 mL of serum-deprived media containing 20 ng/mL of LPS. Twenty microliters of postbiotics or MRS were added immediately. After 2 h of priming, cSiO_2_ suspension was added to reach a final concentration of 25 µg/mL. After 4 h of incubation, supernatants were collected for LDH and cytokine ELISA assays.

### 2.2. Bacterial Strains and Preparation of Postbiotics

The bacterial cultures of LGG and L.reu were kindly gifted by Dr. Steven Goodman from the Ohio State University. BB12 was obtained from Chr. Hansen Inc. (Hørsholkm, Denmark). All the strains were validated by Sanger sequencing and identified to the species level using BLAST (https://blast.ncbi.nlm.nih.gov/Blast.cgi, accessed on 8 July 2019) prior to conducting the experiments. Identity scores of all the strains were ≥ 99%. Stock cultures were maintained at −80 °C with 30% glycerol. The LGG and L.reu cultures were revived by streak plating on DeMan, Rogosa Sharpe (MRS) agar (BD Difco™, Franklin Lakes, NJ, USA) and incubated at 37 °C for 24 h in a digital incubator (VWR, Radnor, PA, USA). A single colony was picked and enriched in MRS broth for 18 h at 37 °C. The BB12 was revived with the same method; however, all the steps were carried out anaerobically. The MRS was supplemented with 0.05% cysteine (Sigma-Aldrich), which was used as a growth medium for BB12. Enriched cultures were collected by centrifugation at 1000× *g* for 10 min. The cultures were washed twice with PBS and diluted with a growth medium to reach an optical density (OD) of 0.1 at 600 nm, which corresponds to approximately 10^8^ CFU/mL. Then, the cultures were incubated under aerobic (LGG and L.reu) or anaerobic (BB12) conditions at 37 °C. When the bacterial cultures reached an OD of 1, 2, 3, 4 or 5 (±0.15), the cell-free fractions (postbiotics) were collected by centrifugation (Eppendorf 5418, Hamburg, Germany) at 1000× *g* for 15 min at 4 °C, followed by filtering the supernatant through a 0.2 µm syringe filter to remove the remaining bacterial cells and debris. The collected postbiotics were aliquoted and stored at −80 °C until use. Only the cell-free fractions were collected from these bacterial cultures and will be referred to as LGG, L.reu, and BB12 for the remainder of the manuscript.

### 2.3. Cell Lines and Culture Conditions

Both the WT and ASC cell lines were obtained as gifts from Dr. James J. Pestka from Michigan State University. The WT and ASC cells were cultured in a phenol red-free RPMI medium (Gibco, Waltham, MA, USA), supplemented with a 10% fetal bovine serum (Gibco) and 1% penicillin–streptomycin (Gibco). Phenol red was not used in the culture medium to avoid potential interferences with downstream colorimetric assays. The cells were incubated in a humid CO_2_ (5%) incubator (VWR) at 37 °C and were sub-cultured every two to three days.

### 2.4. cSiO_2_ Preparation

Crystalline silica (cSiO_2_) (Min-U-Sil-5, Pennsylvania Glass Sand Corp, Pittsburgh, PA, USA) was prepared as described by Wierenga, K.A. et al. [24]. Briefly, cSiO_2_ was suspended in 1 M HCl and incubated at 100 °C for 1 h. After cooling, the acid-washed cSiO_2_ was washed three times in sterile water and dried overnight at 200 °C. Prior to use, the dried cSiO_2_ was suspended in sterile Dulbecco’s phosphate-buffered saline (DPBS, Gibco) at a concentration of 2.5 mg/mL, vortexed for one minute, and sonicated for one minute. The cSiO_2_ suspension was added to the cells in a dropwise manner followed by a gentle manual swirl of the plate to ensure the even distribution of the particles. The cytotoxicity of the cSiO_2_ at the final concentrations of 6.25, 12.5, 25 and 50 µg/mL was tested, and a dose response curve in cytotoxicity was observed when the cSiO_2_ concentrations increased from 6.25 to 25 ug/mL; thus, 25 µg/mL was selected for the following experiments.

### 2.5. Cytotoxicity

The cytotoxicity of the treated cells was measured using an LDH assay as previously described [27]. Briefly, iodotetrazolium chloride (INT, VWR) in DMSO (VWR) at 33 mg/mL, phenazine methosulfate (PMS, Tokyo Chemical Industry Co., Ltd., Tokyo, Japan) in water at 9 mg/mL, NAD (VWR) in water at 8.6 mg/2.3 mL and lithium lactate (Alfa Aesar, Haverhill, MA, USA) in water at 49 mg/2.5 mL were pre-made, aliquoted, and stored at −20 °C. A tris buffer (200 mM, pH 8) was prepared and stored at 4 °C until use. All the reagents were brought to room temperature prior to use. In total, 2% of the detergent (10% Triton X-100, VWR) was added to the control wells to induce maximum lysis. Supernatants from the lysate were collected and used as maximum kill samples (MK). A total of 50 µL of MK and supernatants from the treated cells were loaded to a flat-bottomed, clear 96-well plate. A solution of 200 µL of INT, 200 µL of PMS, and 4.6 mL of NAD was mixed with 5 mL of lithium lactate and 5 mL of tris buffer. Then, 100 µL of the mixture was immediately loaded to each well of the 96-well plate. The plate was incubated in the dark at room temperature for 15 min. Absorbance at 492 nm was measured using a BioTek Epoch 2 plate reader (BioTek Instruments, Inc., Winooski, VT, USA). The cytotoxicity was calculated using the following equation: 100% × [(Sample_absorbance_ − Sample Blank_absorbance_)/(MK_absorbance_ − MK Blank_absorbance_)], where “Sample_absorbance_“ is the absorbance of wells containing supernatants from postbiotics or MRS-treated cells, “Sample Blank_absorbance_” is the absorbance of wells containing serum-deprived RPMI medium instead, “MK_absorbance_“ is the absorbance of wells containing MK lysate, and “MK Blank_absorbance_” is the absorbance of wells containing serum-deprived RPMI with 10% Triton-X.

### 2.6. IL-1 Cytokine Analyses

The released IL-1α and IL-1β from the postbiotics-treated cells were measured using mouse IL-1α/IL-1F1 and IL-1β/IL-1F2 ELISA kits from DuoSet^®^ (R&D Systems, Minneapolis, MN, USA), respectively. The procedures were carried out according to the manufacturer’s instructions.

### 2.7. Statistical Analyses

Comparisons of the impact of the postbiotics collected at different ODs were conducted using a one-way ANOVA followed by Tukey’s HSD test. The Student *t*-tests were used to compare the WT and ASC cells. *p* < 0.05 was considered statistically significant.

## 3. Results

### 3.1. Effect of Postbiotics on Macrophage Survivability

To establish the optimal OD range for bacterial growth and postbiotics treatment within our in vitro model, the cytotoxicity of the postbiotics collected from OD 1 to 5 was measured using an LDH assay. The release of LDH into the cell culture medium was used as an indicator of cell death. Compared to the MRS, the LGG collected at OD 4 and 5 resulted in 2.3- and 3.0-fold increases in the ASC cell death, and 2.0- and 2.6-fold increases in WT, while fold increases caused by the L.reu obtained at OD 4 and 5 were 1.8- and 3.0-fold in ASC, and 1.5- and 2.2-fold in WT, respectively. Treatments with the BB12 at OD 4 and 5 caused 1.5- and 2.0-fold increases in the ASC cell death, whereas OD 4 and 5 caused a 1.3-fold increase in the cell death of WT compared to the control MRS treatment. For both the ASC and WT cells, the postbiotics collected from OD 1, 2, and 3 did not cause cytotoxicity as measured by an LDH assay and, thus, were used for downstream experiments (Figure 2).

### 3.2. Effect of Postbiotics on cSiO_2_-Induced Cytotoxicity in Macrophages

To evaluate the effect of the postbiotics on the cSiO_2_-stimulated macrophages, postbiotics from LGG, L.reu, and BB12 were added 24 h before LPS priming, followed by cSiO_2_ stimulation. We hypothesize that the postbiotics would be protective against LPS-primed cSiO_2_-induced cytotoxicity in the RAW 264.7 macrophage model. We demonstrate that all three candidate postbiotics decreased cSiO_2_-induced cell death in both the ASC and WT cells compared to the control MRS-treated cells, and this effect was OD dependent (Figure 3). The LDH levels released from the LPS-primed cSiO_2_-stimulated ASC cells treated with LGG from all three doses (OD 1–3) significantly decreased LDH levels to 11.1%, 9.8%, and 8.5%, respectively, compared to the MRS control (13.2%). The LDH reduction in the L.reu-treated ASC cells demonstrates a significant reduction in cytotoxicity as follows: 10.0%, 8.6%, and 8.7% when pre-treated with the L.reu postbiotics from OD 1, 2, and 3, respectively. The BB12 collected from OD 1, 2, and 3 resulted in similar decreases in LDH levels to 11.0%, 9.0%, and 7.8%. Thus, similar reductions in cSiO_2_-induced cytotoxicity were observed when the ASC cells were pre-treated with all three postbiotics collected from OD 1–3.

OD-dependent protective effects of the postbiotics were observed in cSiO_2_-induced cytotoxicity of the WT cells. Compared to the MRS (control), the treated WT (12.4% LDH release) postbiotics collected at OD 1, 2, and 3 from LGG reduced LDH levels to 11.8%, 10.2% and 8.8%, respectively. The released LDH levels were reduced to 10.7%, 9.4%, and 8.3% when the WT cells were treated with the L.reu collected at OD 1 to 3. The BB12 collected at OD 1, 2, and 3 reduced LDH levels to 10.2%, 7.9%, and 7.5%, respectively. It appears that the L.reu and BB12 collected at OD 1 demonstrate similar protective effects on cSiO_2_-induced cytotoxicity while treatment with the LGG did not alter the released LDH levels compared to the MRS. At OD 2, the BB12 was most effective at decreasing cytotoxicity compared to the LGG and L.reu. At OD 3, the BB12-treated WT cells released lower levels of LDH compared to the LGG-treated WT, while the L.reu had similar effects to either the BB12 or LGG postbiotics. Thus, we conclude that the postbiotics from LGG, L.reu and BB12 obtained from OD 1-3 exhibited a dose-dependent protective effect on cSiO_2_-induced cytotoxicity of the WT cells.

### 3.3. Postbiotics Suppressed Proinflammatory Cytokine Release

To assess the effect of the postbiotics obtained from LGG, L.reu, and BB12 on the release of IL-1β and IL-1α, three different concentrations (cell-free fractions from OD 1, 2, and 3) were added to the ASC or WT cells after LPS priming (20 ng/mL) and cSiO_2_ stimulation (25 µg/mL). The IL-1 cytokine release was measured in culture supernatants (Figure 4). The MRS control was used as a positive control in all the experiments. The postbiotics collected from LGG, L.reu, and BB12 suppressed the release of IL-1β (*p* < 0.05) in the ASC, but not the WT cells. However, the observed reduction does not appear to be OD-dependent. For example, the LGG obtained from OD 1, 2, and 3 decreased the IL-1β levels to 141.3 pg/mL, 122.6 pg/mL, and 119.7 pg/mL compared to the MRS control (272.6 pg/mL). In addition, the L.reu and BB12 addition to the ASC cells resulted in similar reductions in the IL-1β release. We observed a 53–60% decrease in the IL-1β levels released by the ASC cells treated with postbiotics, suggesting a protective role against LPS priming and cSiO_2_ stimulation of inflammatory endpoints. The most significant decreases in the IL-1β levels were observed by the BB12 followed by the L.reu and LGG. A three-fold increase in the magnitude of the IL-1β release was observed in the ASC cells compared to WT (Figure 4A–C). When the ASC and WT cells were treated with the MRS control, the cSiO_2_-induced release of IL-1β was significantly higher in ASC (272.7 pg/mL) compared to WT (70.7 pg/mL).

The postbiotics collected from LGG, L.reu, and BB12 did not alter the release of IL-1α relative to IL-1β in both the ASC and WT cells. Compared to the MRS control, the LGG at all concentrations did not alter the IL-1α levels in the ASC and WT cells. We observed a slight reduction in the IL-1α responses in the ASC cells pre-treated with the BB12 at OD 3, but not at OD 1 and 2 (Figure 4E). Compared to the control, the L.reu postbiotics collected at OD 1 and 2 slightly enhanced the release of IL-1α by the WT cells after LPS priming and cSiO_2_ stimulation, but the WT cells treated with the L.reu postbiotics collected at OD 3 released a similar amount of IL-1α as control. Taken together, we conclude that the presence of a functional inflammasome in the ASC cell model is necessary for a robust IL-1β cytokine response upon LPS priming and cSiO_2_ stimulation. In addition, the reduction in the IL-1β level as a result of the postbiotics pre-treatment in the ASC cells appears to be OD-independent. Finally, the mechanism of IL-1β release is likely via a different pathway compared to the secretion of IL-1α.

## 4. Discussion

Here, we used a previously established in vitro RAW 264.7 macrophage model to evaluate the effects of postbiotics obtained from LGG, L.reu, and BB12 on cSiO_2_-induced cytotoxicity, and the subsequent impact on IL-1 cytokines secretion. We first tested the cytotoxicity of postbiotics on macrophages without cSiO_2_ stimulation to rule out the reduction in cell numbers and inflammatory cytokines from postbiotics in downstream experiments. It is necessary to determine the cytotoxicity of bioactive compounds before evaluating their potential benefits using in vitro cell models because bioactive compounds may exert a toxic effect on cells even at a normal physiological concentration [28]. For example, butyrate, a short-chain fatty acid (SCFA) produced by probiotic bacteria was reported to induce cell death in epithelial cells [29]. In our study, we found that postbiotics from LGG, L.reu, and BB12 collected at OD 1 to 3 did not induce cytotoxicity in macrophages; thus, they were utilized in downstream experiments.

The postbiotics obtained from LGG, L.reu, and BB12 decreased the LDH levels from cSiO_2_-treated macrophages, indicating that postbiotics could protect macrophages from cSiO_2_-induced cell death. Macrophage cell death leads to the release of proinflammatory cytokines that could aggravate chronic inflammation [30]. By protecting macrophages from cSiO_2_-induced death, postbiotics may contribute to ameliorate cSiO_2_-triggered inflammatory responses via reducing the release of proinflammatory molecules from dead macrophages. The protective effects of postbiotics on macrophages were OD-dependent, which means the effects were dependent on the time when the postbiotics were collected from the culture broth of the probiotics. Similar findings were reported by Cicenia A. et al., in that the protective effect of postbiotics from LGG on the LPS-induced impairment of human colonic smooth muscle cells depended on the collection time of growth [31]. In addition, the authors suggested that the different effects of postbiotics collected at varied growth phases could be related to the presence/absence of bioactive mediators in postbiotics at different growth phases [31].

Compared to the WT cells, the ASC cells produced almost three times higher levels of IL-1β compared to the MRS control. Since the release of IL-1β from cSiO_2_-stimulated macrophages involves the assembly of the NLRP3 inflammasome [32], this observation indicates that ASC cells are a more responsive and robust model than WT cells to investigate molecular mechanisms that involve NLRP3 inflammasome activation. Relative to IL-1α release, we did not observe a consistent pattern between the ASC and WT cells when they were treated with the same postbiotics. This result may be explained by a conclusion drawn by Gross, O., et al., in that the cSiO_2_-induced release of IL-1α could be NLRP3 inflammasome-dependent or independent in bone-marrow-derived dendritic cells [33]. We also observed that postbiotics from LGG, L.reu, and BB12 significantly reduced the release of mature IL-1β by LPS-primed, cSiO_2_-stimulated macrophages, which suggests that these postbiotics exhibit anti-inflammatory activity in vitro and may contribute to reducing chronic inflammation in vivo. Short-chain fatty acids are compounds of interest produced by postbiotics that may contribute to the observed reduction in IL-1 cytokine release. Other studies confirmed that treating macrophages with simple SCFA decreased the release of pro-inflammatory cytokines, including IL-1 in an NF-ƙB-dependent mechanism [34,35]. SCFA are not the only set of bioactive compounds in postbiotics that are proposed to exert anti-inflammatory activities; other compounds, such as exopolysaccharides, amino acids, and proteins [36], could also help to explain the observed reduction in cytotoxicity and IL-1 release in this study. Follow-up studies will profile metabolites, including SCFA, exopolysaccharides, and amino acids, in postbiotics obtained from LGG, L.reu, and BB12 in a time-dependent manner.

## 5. Conclusions

In this study, we demonstrated that postbiotics from LGG, L.reu, and BB12 decreased cSiO_2_-induced cytotoxicity in RAW 264.7 murine macrophages in vitro, and reduced the release of IL-1β, a pro-inflammatory cytokine, from LPS-primed macrophages after cSiO_2_ stimulation. These results indicate that postbiotics from LGG, L.reu, and BB12 exhibit anti-inflammatory activity and may contribute to the amelioration of cSiO_2_-induced inflammation. In addition, we compared the responses from two macrophage models (WT and ASC) upon cSiO_2_ stimulation and conclude that ASC cells (wild-type RAW264.7 cells transfected with ASC proteins) are more responsive to cSiO_2_ triggering than WT cells, and provide a better model to study inflammatory responses of macrophages that may involve activation of the NLRP3 inflammasome.

## Figures and Tables

**Figure 1 foods-11-00877-f001:**
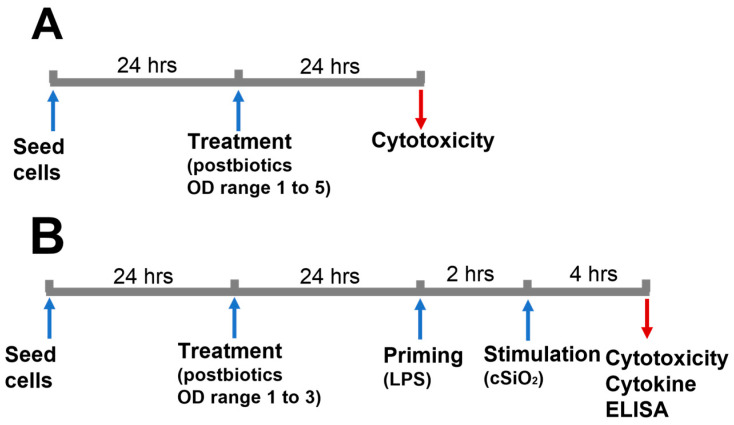
Experimental design. The experiment aims to: (**A**) determine the OD range to collect postbiotics that do not affect the cell viability for downstream experiments; and (**B**) investigate the impacts of postbiotics on cSiO_2_-induced cytotoxicity and the release of IL-1 cytokines of macrophages. The upward-facing arrows indicate the addition of cells and reagents, while the downward-facing arrows indicate the removal of samples for analyses.

**Figure 2 foods-11-00877-f002:**
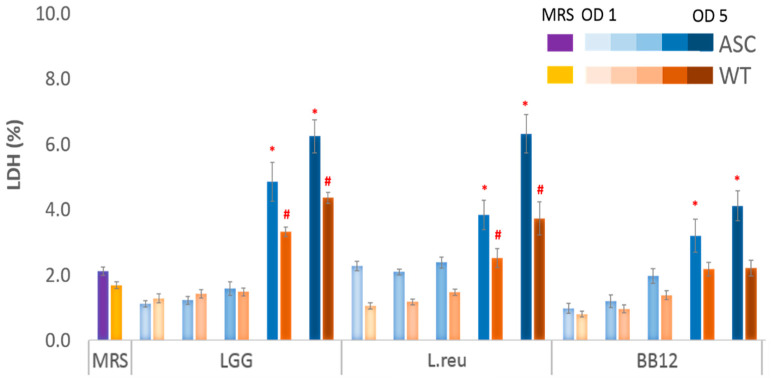
Effects of the postbiotics collected at OD 1 to 5 on the cell viability. The postbiotics collected from OD 1 to OD 3 did not result in cytotoxicity of the WT or ASC cells; thus, they were used in the following experiments. The postbiotics at OD 4 and 5 resulted in cytotoxicity of the WT and ASC cells, and the magnitude of cytotoxicity was higher in the ASC cells. Assays were conducted after treating the cells with postbiotics for 24 h, without LPS priming and cSiO_2_ stimulation. The data are presented as mean ± SEM; the results were pooled from the biological triplicate. The bars marked with * are significantly higher than the MRS-treated ASC cells (*p* < 0.05); the bars marked with # are significantly higher than the MRS-treated WT cells (*p* < 0.05).

**Figure 3 foods-11-00877-f003:**
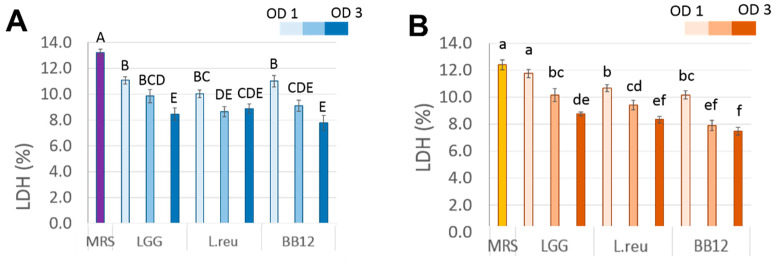
Postbiotics exhibit OD-dependent protective effects on cSiO_2_-induced cell death in: (**A**) the ASC cells; and (**B**) the WT cells after priming with LPS. The data are presented as mean ± SEM; biological triplicates were conducted. The capitalized letters indicate significant differences among ASC, while different lowercase letters indicate significant differences among WT (*p* < 0.05).

**Figure 4 foods-11-00877-f004:**
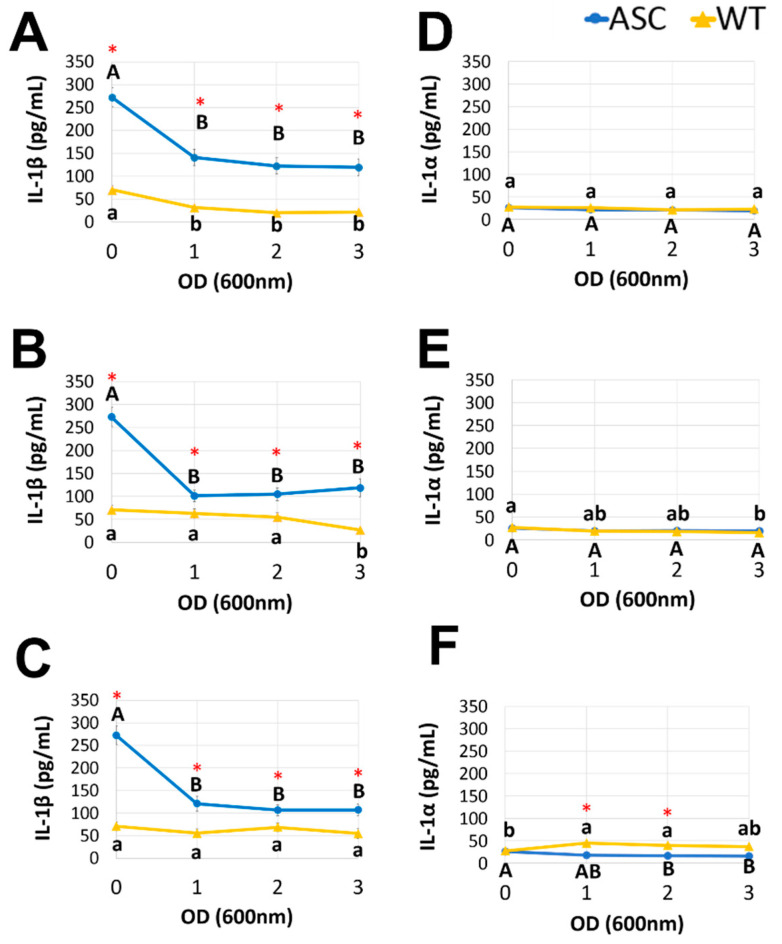
Postbiotics collected from LGG, L.reu, and BB12 suppressed the release of IL-1β in the ASC, but not the WT cells. The cytokines, IL-1β (panels **A**–**C**), and IL-1α (panels **D**–**F**) were analyzed by ELISA in supernatants collected from the ASC and WT cells after LPS priming (20 ng/mL) and cSiO_2_ stimulation (25 µg/mL). Prior to the LPS priming and cSiO_2_ stimulation, 1 × 10^5^ cells/well were pre-treated with postbiotics collected from OD 1–3 of LGG (panels **A**,**D**), BB12 (panels **B**,**E**), and L.reu (panels **C**,**F**). The data are presented as mean ± SEM; biological triplicates were conducted. The cells treated with the MRS only were used as a negative control for postbiotic pre-treatment (designated data point OD 0). The blue lines with circles represent the ASC cells and the yellow lines with triangles represent the WT cells. The capitalized letters indicate significant differences among ASC while the lowercase letters indicate significant differences among WT (*p* < 0.05). * Indicates significant differences between ASC and WT (*p* < 0.05).

## Data Availability

Data is contained within the article.

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
