# Peer review of "Dietary Postbiotics Reduce Cytotoxicity and Inflammation Induced by Crystalline Silica in an In Vitro RAW 264.7 Macrophage Model"

_foods, 2022, doi:10.3390/foods11060877_

Round 1
Reviewer 1 Report
The manuscript is well written and the experiments are well designed and conducted. I only have a few comments:
- Line 14: there is no need to write “Bb12” with italics.
- The term "microflora" is incorrect when referring to GIT microorganisms, as we will not find plants there. Authors should use the terms "microbiota" or "microbiome" depending on the context.
- Introduction: the sentence “Crystalline silica (cSiO2) are naturally existing particles that are environmental toxicants to humans and animals” – the authors could list the sources of these pollutants. What are these sources? Where do they come from in the environment? What is the exposure for humans?
- Emphasise the novelty of the study.
- Did you neutralize the pH of cell free culture supernatants? What was the pH of postbiotics used? Put it in the text. It is known that acidic pH can cause cytotoxicity and cell death, how will the authors interpret their results in terms of pH?
Author Response
Thank you for your time in reviewing our manuscript and providing constructive feedback to help improve the final manuscript. Attached are point-by-point responses, also available in full as attached PDF.
- Line 14: there is no need to write “Bb12” with italics.
Response: Font has been changed to the default font.
- The term "microflora" is incorrect when referring to GIT microorganisms, as we will not find plants there. Authors should use the terms "microbiota" or "microbiome" depending on the context.
Response: Thank you for pointing this out: “Microflora” in L59 has been changed to “microbiota”.
- Introduction: the sentence “Crystalline silica (cSiO2) are naturally existing particles that are environmental toxicants to humans and animals” – the authors could list the sources of these pollutants. What are these sources? Where do they come from in the environment? What is the exposure for humans?
Response: In L66-69, a new sentence “Almost every mineral deposit contains cSiO2, and thus, workers in heavy construction industries are exposed to high amounts of cSiO2, for example individuals working in masonry and plastering, heavy construction, painting and paper hanging, iron and steel foundries, and metal services [17].” and a new reference #17 has been added.
- Emphasis the novelty of the study.
Response: In L93-95, a sentence “To our knowledge, this is the first study that utilizes the RAW-ASC model that investigates the effect of dietary postbiotics on cSiO2-induced toxicity and inflammation in-vitro.” has been added to emphasize the novelty.
- Did you neutralize the pH of cell free culture supernatants? What was the pH of postbiotics used? Put it in the text. It is known that acidic pH can cause cytotoxicity and cell death, how will the authors interpret their results in terms of pH?
Response: Thank you for pointing out this and we added a new sentence in Line 110 to provide clarification to this question. We did not neutralize the pH of the cell-free bacteria supernatants, but we measured pH of the mixture of individual postbiotics and serum-deprived RPMI medium using the same postbiotics to medium ratio (1:50) as applied in the experiment, and the pH (7.3 ± 0.1) of the mixtures were not different from serum-deprived RPMI medium alone. Thus, we concluded that after adding postbiotics, the pH of medium in each well did not change and should not affect cell performance.
The new sentence added is “Prior to conducting the experiment, we measured the pH of mixtures containing 40 µL of each postbiotics and 2 mL of serum-deprived medium (same ratio of postbiotics to medium as used in the experiment) to rule out the impact of pH change on cell viability. The pH of individual mixture was 7.3 (± 0.1), which was the same as serum-deprived medium alone.” See L126-130.
Reviewer 2 Report
An up-to-date topic was addressed on the paper, indicating i) cSiO2-induced cytotoxicity reduction exerted by postibiotic effect of lactic acid bacteria, ii) dose-dependent protective effect on cSiO2-induced cytotoxicity. I point out minor concerns to this paper. For guidance:
Update nomenclature ‘Lactobacillus rhamnosus’ with ‘Lacticasebacillus rhamnosus’ and Lactobacillus reuteri with “Limosilactobacillus reuteri’, accordingly to ISAPP
Revise terms introduced by abbreviations in the narrative
Replace ‘microflora’ with ‘microbiota’ or ‘microbiome’
Revise equipment description
I suggest the Authors introduce ‘Material and Methods section’ with topic 2.4 -Experimental Design
Define the terms of the Equation used for cytotoxicity measurements
Replace section titles ‘Postbiotics Collected at OD 1 to 3 Did Not Affect the Viability of Macrophages’ with ‘Effect of Postbiotic on Machophage Survivability’; ‘Postbiotics Reduced the cSiO2-induced Cytotoxicity to Macrophages’ with ‘Effect of Postbiotic on cSiO2-induced Cytotoxicity to Macrophages’
Author Response
Thank you for your time in reviewing our manuscript and providing constructive feedback to help improve the final manuscript. Attached are point-by-point responses, also available in full as attached PDF.
- Update nomenclature ‘Lactobacillus rhamnosus’ with ‘Lacticasebacillus rhamnosus’ and Lactobacillus reuteriwith “Limosilactobacillus reuteri’, accordingly to ISAPP
Response: Thank you for pointing out this. Changes have been made consistently throughout the text in L13, L84-85, and L100-102.
- Revise terms introduced by abbreviations in the narrative
Response: Full names for TNF-alpha and IL-10 and IL-12 cytokines have been added to abbreviations in L53-55, L77-82. Please clarify if this comment has been addressed based on our interpretation of reviewers’ request.
- Replace ‘microflora’ with ‘microbiota’ or ‘microbiome’
Response: “Microflora” on Line 53 has been replaced with “microbiota”.
- Revise equipment description
Response: Information has been added to Line 166-167, L175-176, and L188-189.
- I suggest the Authors introduce ‘Material and Methods section’ with topic 2.4 -Experimental Design
Response: Sub-section “2.4-Experimental design” has been moved to the beginning of “Material and Methods” section, and the title number has been changed to “2.1”. To match this change, the following changes have been made: 1) Sub-section numbers within “Material and Methods” has been changed accordingly; 2) Information (full name and abbreviation) of bacteria strains was moved to L100-101 since L100 is where it first appears in the “Materials and Methods” section; 3) Cell line information was added in L117 for the same reason as bacteria strains; 4) Abbreviation of bacteria strains instead of full name was used in Line 149-151; 5) Introduction of the two cell lines were changed in Line 185 to avoid repetition.
- Define the terms of the Equation used for cytotoxicity measurements
Response: Sentences have been added to L235-239 after the equation to define the terms – “where “Sampleabsorbance“ is the absorbance of wells containing supernatants from postbiotics or MRS treated cells, “Sample Blankabsorbance” is the absorbance of wells containing serum-deprived RPMI medium instead, “MKabsorbance“ is the absorbance of wells containing MK lysate and “MK Blankabsorbance” is the absorbance of wells containing serum-deprived RPMI with 10% Triton-X.”
- Replace section titles ‘Postbiotics Collected at OD 1 to 3 Did Not Affect the Viability of Macrophages’ with ‘Effect of Postbiotic on Machophage Survivability’; ‘Postbiotics Reduced the cSiO2-induced Cytotoxicity to Macrophages’ with ‘Effect of Postbiotic on cSiO2-induced Cytotoxicity to Macrophages’
Response: Great suggestions, thank you! Changes have been made to these titles accordingly.

Reviewer 3 Report
This manuscript is very well written. The research about postbiotics is very interesting and novelty, the interest of the readers it will be high. However, some comments should be adressed.
- Although that ISAAP given a re-definition of the term postbiotic, other scientists experts have pointed to some inconsistencies and ambiguities associated with the recently proposed re-definition of postbiotics (Nat Rev Gastroenterol Hepatol 18, 825–826 (2021). https://doi.org/10.1038/s41575-021-00521-6). In my opinion, the postbiotics tested by the authors is in agreement with the original defintion given by the references 2 and 12 of your manuscript. The authors can revise the following recent articles (Food Research International 136. 109502. https://doi.org/10.1016/j.foodres.2020.109502; Nat Rev Gastroenterol Hepatol 18, 825–826 (2021). https://doi.org/10.1038/s41575-021-00521-6). I strongly suggest to the authors modified accordingly the definition of postbiotics and add some additional information about this debated term.
- Please use the new taxonomy for Lactobacillus genus described in the manuscript. For example, Lactobacillus reuteri is now called as Limosilactibacillus reuteri and Lactobacillus rhamnosus is now called as Lacticaseibacillus rhamnosus. The authors can revise the following ISAPP´s infographic (https://4cau4jsaler1zglkq3wnmje1-wpengine.netdna-ssl.com/wp-content/uploads/2020/04/Lactobacillus_consumer_2020-1.pdf).
- Line 327-L332. There are more information about of bioactive compounds in postbiotics with proposed anti-inflammatory activities. For example, in the following article (Probiotics and Antimicrobial Proteins (2020) 12:608–622. https://doi.org/10.1007/s12602-019-09568-z) are described some amino acids in postbiotics with anti-inflammatory activity. Please added the information.
Author Response
Thank you for your time in reviewing our manuscript and providing constructive feedback to help improve the final manuscript. Attached are point-by-point responses, also available in full as attached PDF.
. Although that ISAAP given a re-definition of the term postbiotic, other scientists experts have pointed to some inconsistencies and ambiguities associated with the recently proposed re-definition of postbiotics (Nat Rev Gastroenterol Hepatol 18, 825–826 (2021). https://doi.org/10.1038/s41575-021-00521-6). In my opinion, the postbiotics tested by the authors is in agreement with the original defintion given by the references 2 and 12 of your manuscript. The authors can revise the following recent articles (Food Research International 136. 109502. https://doi.org/10.1016/j.foodres.2020.109502; Nat Rev Gastroenterol Hepatol 18, 825–826 (2021). https://doi.org/10.1038/s41575-021-00521-6). I strongly suggest to the authors modified accordingly the definition of postbiotics and add some additional information about this debated term.
Response: Thank you for pointing this out. Sentences and new references have been added in L32-37 to add the previous definition and existing debate between the two definitions. Sentences are “The definition of postbiotics has been previously defined as “any factor resulting from the metabolic activity of a probiotic or any released molecule capable of conferring beneficial effects to the host in a direct or indirect way” [2]. Though the new definition by ISAPP categorizes inactivated microorganisms as postbiotics, most studies about postbiotics focus on the metabolites from probiotic microorganisms, not the inactivated microorganisms [3].
- Please use the new taxonomy for Lactobacillus genus described in the manuscript. For example, Lactobacillus reuteri is now called as Limosilactibacillus reuteri and Lactobacillus rhamnosus is now called as Lacticaseibacillus rhamnosus. The authors can revise the following ISAPP´s infographic (https://4cau4jsaler1zglkq3wnmje1-wpengine.netdna-ssl.com/wp-content/uploads/2020/04/Lactobacillus_consumer_2020-1.pdf).
Response: Changes have been made in L13 and all instances in the manuscript. Thank you for pointing this out.
- Line 327-L332. There are more information about of bioactive compounds in postbiotics with proposed anti-inflammatory activities. For example, in the following article (Probiotics and Antimicrobial Proteins (2020) 12:608–622. https://doi.org/10.1007/s12602-019-09568-z) are described some amino acids in postbiotics with anti-inflammatory activity. Please added the information.
Response: The suggested information has been added to L399-402 with a new reference #36 recommended by reviewer. The sentence now reads, “SCFA are not the only set of bioactive compounds in postbiotics that have been proposed to exert anti-inflammatory activities, other compounds, such as exopolysaccharides, amino acids and proteins [36], could also help explain observed reduction in cytotoxicity and IL-1 release in this study.”
